

# Gene expression profiles of Japanese precious coral *Corallium japonicum* during gametogenesis

Ma. Marivic Capitle Pepino[1], Sam Edward Manalili[1], Satoko Sekida[2], Takuma Mezaki[3], Tomoyo Okumura[4] and Satoshi Kubota[2]

[1] Kuroshio Science Program, Graduate School of Integrated Arts and Sciences, Kochi University, Nankoku, Kochi, Japan
[2] Kuroshio Science Unit, Multidisciplinary Science Cluster, Kochi University, Nankoku, Kochi, Japan
[3] Kuroshio Biological Research Foundation, Otsuki, Kochi, Japan
[4] Marine Core Research Institute, Kochi University, Nankoku, Kochi, Japan

Corresponding author
Ma. Marivic Capitle Pepino,
marivicpepino@gmail.com

## ABSTRACT

**Background**. *Corallium japonicum*, a prized resource in Japan, plays a vital role in traditional arts and fishing industries. Because of diminished stock due to overexploitation, ongoing efforts are focused on restoration through transplantation. This study aimed to enhance our understanding of the reproductive biology of these valuable corals and find more efficient methods for sex determination, which may significantly contribute to conservation initiatives.

**Methods**. We used 12 three-month aquarium reared *C. japonicum* colony fragments, conducted histological analysis for maturity and sex verification, and performed transcriptome analysis via *de novo* assembly and mapping using the *C. rubrum* transcriptome to explore gene expression differences between female and male *C. japonicum*.

**Results**. Our histological observations enabled sex identification in 33% of incompletely mature samples. However, the sex of the remaining 67% of samples, classified as immature, could not be identified. RNA-seq yielded approximately 21–31 million short reads from 12 samples. *De novo* assembly yielded 404,439 highly expressed transcripts. Among them, 855 showed significant differential expression, with 786 differentially expressed transcripts between females and males. Heatmap analysis highlighted 283 female-specific and 525 male-specific upregulated transcripts. Transcriptome assembly mapped to *C. rubrum* yielded 28,092 contigs, leading to the identification of 190 highly differentially expressed genes, with 113 upregulated exclusively in females and 70 upregulated exclusively in males. Blastp analysis provided putative protein annotations for 83 female and 72 male transcripts. Annotation analysis revealed that female biological processes were related to oocyte proliferation and reproduction, whereas those in males were associated with cell adhesion.

**Discussion**. Transcriptome analysis revealed sex-specific gene upregulation in incompletely mature *C. japonicum* and shared transcripts with *C. rubrum*, providing insight into its gene expression patterns. This study highlights the importance of using both *de novo* and reference-based assembly methods. Functional enrichment analysis showed that females exhibited enrichment in cell proliferation and reproduction pathways, while males exhibited enrichment in cell adhesion pathways. To the best of our knowledge, this is the first report on the gene expressions of each sex during the spawning season. Our findings offer valuable insights into the physiological ecology of

incompletely mature red Japanese precious corals and suggest a method for identifying sex using various genes expressed in female and male individuals. In the future, techniques such as transplantation, artificial fertilization, and larval rearing may involve sex determination methods based on differences in gene expression to help conserve precious coral resources and ecosystems.

## INTRODUCTION

Precious corals, belonging to the class Octocorallia (Order Scleralcyonacea) (*McFadden, Van Ofwegen & Quattrini, 2022*), are slow-growing, long-lived organisms composed of calcium carbonate (*Iwasaki & Suzuki, 2010*). They are used to make jewelry and other crafts in several harvesting areas, thereby supporting local fishing and traditional arts industries (*Nonaka & Muzik, 2009*). The Mediterranean Sea and Northern Pacific Ocean (Japan, Taiwan, and China) are two major regions with large populations that are commercially exploited (*Cannas et al., 2019*). Fishery stocks are currently in limited supply in both regions owing to their high demand, and active restoration efforts have been suggested (*Tsounis et al., 2010*). The most significant advancement in precious coral restoration over the last decade has been transplantation using coral fragments (*Koido et al., 2022*; *Villechanoux et al., 2022*).

Research on transplantation, artificial fertilization, and larval culture has been advanced in scleractinian corals of the class Hexacorallia (*Omori & Iwao, 2014*; *Leal et al., 2016*; *Lam et al., 2023*). These procedures involve collecting gravid coral colonies, rearing them in land-based aquaria, and allowing them to spawn, fertilize, and raise larvae (*Omori & Iwao, 2014*). This represents the most advanced approach for the efficient and effective restoration of coral populations. These techniques can also be used to increase the populations of precious corals.

To implement these techniques successfully, accurate information regarding the timing of coral spawning, age, size at first maturity, and sex identification is essential. However, limited studies have been conducted on the reproductive biology of precious corals (*Torrents et al., 2005*; *Torrents & Garrabou, 2011*; *Nonaka, Nakamura & Muzik, 2015*; *Santangelo et al., 2003*; *Sekida, Iwasaki & Okuda, 2016*). Based on year-round observations of gonad maturity, Japanese precious coral species spawn from May to August (*Nonaka, Nakamura & Muzik, 2015*; *Sekida, Iwasaki & Okuda, 2016*), whereas Mediterranean precious coral species spawn in July and August (*Santangelo et al., 2003*).

The age and size at which precious corals reach reproductive maturity vary significantly among coral species. In the case of Japanese precious corals, specific size information on reproductive maturity is lacking. *Nonaka, Nakamura & Muzik (2015)* provided estimates, suggesting that the minimum size for first maturity is smaller than the smallest fertile

samples observed in their research. Their findings indicated minimum sizes of 145 mm, 223 mm and 78 mm for *Corallium japonicum*, *Pleurocorallium elatius*, and *Pleurocorallium konojoi*, respectively, with corresponding estimates for the age at maturity. In contrast, the Mediterranean red coral, *Corallium rubrum*, exhibits even smaller fertile colonies and a younger age at first maturity (*Santangelo et al., 2003*; *Torrents et al., 2005*). This introductory exploration of age and size at first maturity sets the stage for deeper examination of the reproductive biology of various precious coral species. Several of these species, including *C. japonicum* (*Nonaka, Nakamura & Muzik, 2015*; *Sekida, Iwasaki & Okuda, 2016*), *P. elatius* (*Nonaka, Nakamura & Muzik, 2015*), *C. rubrum* (*Tsounis et al., 2006*), *C. secundum*, and *C. lauuense* (*Waller & Baco, 2007*), have been reported to exhibit a bi-annual oocyte reproductive cycle. This cycle involves two size classes of oocytes: one remaining immature as a reservoir throughout the year, and the other maturing for the next annual spawning. In contrast, sperm cysts consistently follow the annual reproductive cycle.

Moreover, no method has been reported for identifying the sex of precious corals based on morphological characteristics (*i.e.*, color, polyp structure, or skeletal structure); thus, only histological methodology has been applied for sex identification (*Sekida, Iwasaki & Okuda, 2016*). Transcriptome analysis is a promising method of identifying sex because it involves RNA sequencing (RNA-seq), which is a powerful technique for profiling gene expression. Researchers can compare the gene expression profiles between female and male corals. Bioinformatic tools can be used to identify genes that show significant differences in expression between the two sexes. Genes that are consistently up- or downregulated in one sex can serve as markers for sex determination.

Transcriptome analysis has been used to examine gene expression in scleractinian corals and compare female and male gene expression patterns. For example, in the gonochoric hexacoral *Euphyllia ancora* (*Chiu et al., 2020*), the gene expression patterns of oocyte and sperm were compared across various stages of gonad development, revealing sex-specific upregulated genes. The main objective of the transcriptome assembly was to provide a valuable tool for future advancements in identifying germ cell markers specific to either sex or developmental stage. These resources will greatly assist coral aquaculture and ecological research. In octocorals, transcriptome analyses have been performed in *Heliopora, Plexaura, Calyptrophora, Chrysogorgia,* and *Eleutherobia* (*Guzman et al., 2018*; *Pelosi et al., 2022*; *Ryu et al., 2019*; *Ryu, Hwang & Woo, 2023*), as well as in the Mediterranean red precious coral *Corallium rubrum* (*Pratlong et al., 2015*). However, these studies aimed to evaluate the stress response, provide baseline data, and clarify the relationship between common species, but have not compared gene expression between females and males (*Guzman et al., 2018*; *Pelosi et al., 2022*; *Pratlong et al., 2015*; *Ryu et al., 2019*; *Ryu, Hwang & Woo, 2023*).

In this study, we employed two transcriptome analysis methods: *de novo* assembly and reference-based assembly. *De novo* assembly is independent but may result in fragmented transcripts, whereas reference-based assembly uses references but may overlook species-specific genes. Our research focused on the red Japanese precious coral *Corallium japonicum* to demonstrate differences in gene expression between females and males during the spawning season. Our findings offer insights into its physiological ecology and suggest a

way to distinguish between the sexes using a variety of genes expressed in female and male individuals.

## MATERIALS & METHODS

### Sampling and RNA extraction

The red coral colonies used in this study were initially collected by fishermen at an approximate depth of 100 m off Cape Ashizuri (Kochi, Japan) through trawling. We acquired small coral fragments, approximately 2.0–10.0 cm long, with reduced commercial value. Because these fragments were already separated from their original colonies, the specific source colony and the maturity stage of each fragment was unknown. The corals were reared in an aquarium tank at the Institute (Kuroshio Biological Research Foundation). Seawater was sourced locally and constantly cycled into aquarium tanks. The water temperature in the aquarium tanks was maintained at approximately 14–18 °C, and the colonies were fed mixed food two to four times a week. The aquarium tanks were covered with a black opaque panel to reproduce *in situ* light conditions. The colonies were reared in the tank for approximately three months before coral nubbins were collected for the experiments.

In the month prior to the experiment, a preliminary investigation was conducted in the previous month, which allowed us to determine the approximate size of the nubbins to be cut and the portions that were best for RNA-Seq analysis. Based on preliminary data, nubbins cut from the tip yielded higher concentrations of RNA. However, based on the results of *Sekida, Iwasaki & Okuda (2016)*, nubbins cut from this portion tended to be immature. To resolve this problem, we also used nubbins cut from the base, because they have a higher chance of maturation. Additionally, part of the nubbin needed to be histologically observed for sex identification; thus, the middle portion of the coral fragment was used for histological analysis.

Coral nubbins for the experiments were collected on June 4, 2022, considering the reported spawning period from May to August (*Nonaka, Nakamura & Muzik, 2015*; *Sekida, Iwasaki & Okuda, 2016*) and based on observations of the previous year's spawning of cultured precious corals in the Institute tank (June 3–7, 2021; https://www.coral-npo.jp/wp/wp-content/uploads/2022/04/8dc48c7de8c60a3acb78e371ad37ddbb.pdf). Twelve coral fragments were collected, assuming that these were already mature individuals and that we had sufficient replicates for each sex. The apical portion of each coral fragment was cut and divided into three sections (tip, middle, and base). Both tip and basal nubbins were approximately 1–3 cm in length. They were submerged in TRIzol Reagent (Invitrogen, Carlsbad, CA, USA) in cryotubes, snap-frozen in liquid nitrogen, and stored at −80 °C. The middle sections were 2–3 cm long and were fixed in 10% buffered formalin–seawater solution in 15 ml polypropylene tubes.

### Histological observation

Histological slides were prepared as described by *Sekida, Iwasaki & Okuda (2016)*. Gonads were observed under an Olympus SZX16 stereomicroscope (Olympus Co. Ltd., Japan) to identify mature and immature individuals. When gonad-related structures could be

observed under a stereomicroscope, they were considered mature. The gametes were considered immature if they were too small to be observed under a stereomicroscope. Subsequently, the sex was confirmed. Ten ultra-thin sections with a thickness of 50–80 nm were prepared for transmission electron microscopy (TEM) using a Jeol 1010T electron microscope (JEOL Co. Ltd., Japan) to further examine the gonads on a fine scale if they were too difficult to identify using a stereomicroscope. The gonad diameter in sex-identified individuals was measured using ImageJ 1.53t (National Institutes of Health, USA; *Schneider, Rasband & Eliceiri, 2012*), and the average size of oocytes or sperm cysts per individual was calculated.

## RNA extraction, library preparation, and sequencing

The frozen samples were thawed, and the soft tissues of the corals were lysed and homogenized in TRIzol solution. Phase separation, binding, washing, and elution of RNA were performed using the PureLink® RNA Mini Kit (Thermo Fisher Scientific, Waltham, MA, USA), following the manufacturer's protocol. Extracted RNA was checked for occurrence and quality using a NanoVue™ Plus Spectrophotometer (GE Life Sciences, Piscataway, NJ, USA), following the manufacturer's instructions. RNA integrity was confirmed using the Agilent RNA 6000 Nano Kit and Agilent 2100 Bioanalyzer (Agilent Technologies, Santa Clara, CA, USA). Information on the concentration and RNA integrity number (RIN) were summarized and submitted to the sequencing company (GenomeRead Inc., Kagawa, Japan), with 50 μL of each sample. All samples were analyzed regardless of their concentration and RIN to ensure sufficient replication for each coral sex.

RNA samples were prepared using a KAPA mRNA Capture Kit (Cat. KK8440; Kapa Biosystems, Wilmington, MA, USA) and MGIEasy RNA Directional Library Prep Set (Cat. 1000006385, MGI Tech Co., Ltd., Shenzhen, Guangdong, China), according to the manufacturer's protocol. The protocol was initiated by checking the RNA quality using a Qubit Fluorometer (Thermo Fisher Scientific). mRNA was purified using the PolyA method. This was followed by RNA fragmentation to an insert size of 250 bp at 80 °C for 6 min, followed by reverse transcription and second-strand synthesis. Ligation was performed by diluting adapters to 1/10. Fourteen PCR cycles were performed for all 12 samples, followed by purification and quality checks. Sequencing was performed using the DNBSEQ-G400RS High-throughput Sequencing Set (FCL PE150, MGI Tech Co.) using DNBSEQ™ technology.

## Transcriptome analyses

The raw read data received were deposited in the Sequence Read Archive (https://www.ncbi.nlm.nih.gov/sra) under accession number SRA: PRJNA985876. To gain a comprehensive understanding of the transcriptome of *C. japonicum*, two approaches were employed: *de novo* assembly and assembly by mapping against a reference *C. rubrum* transcriptome.

### *De novo* transcriptome assembly of *C. japonicum*

*De novo* transcriptome assembly, annotation, and differential expression analysis were performed following the pipeline provided in Galaxy Training Materials

(https://training.galaxyproject.org/training-material/topics/transcriptomics/tutorials/full-de-novo/tutorial.html) with some modifications. Raw RNA-Seq reads were subjected to quality checks using FastQC v0.12.1. Low-quality reads, adapters, and contaminants were removed to enhance the quality. Fastp and Trimmomatic v0.39 were used to eliminate adapter sequences and low-quality reads. Before transcriptome assembly, another quality control step was performed using FastQC v0.12.1. The *de novo* assembly of the *C. japonicum* transcriptome was performed using Trinity RNAseq v2.15.1, configured with parameters such as–seqType fq,–max memory 60G, and–CPU 24. The 'align_and_estimate_abundance.pl' tool was employed to align reads and estimate transcript abundances, using Bowtie2 and RSEM.

An expression matrix was then generated from estimated abundances using 'abundance_estimates_to_matrix.pl'. This matrix contains estimated counts for each transcript and sample and serves as the input for downstream analysis. Both isoform- and gene-level abundances were processed in this step. Differential gene expression analysis used the 'run_DE_analysis.pl' script and the edgeR method to identify genes or transcripts exhibiting differential expression among various sample groups, namely, females, males, and immature specimens. Low-expression transcripts were filtered out using 'filter_low_expr_transcripts.pl' to retain high-quality transcripts with a minimum expression of one transcript per million. Differential expression analysis was performed by remapping the RNA-Seq reads to the filtered transcriptome. The 'align_and_estimate_abundance.pl' tool was used to align paired-end data to "transcriptome_filtered.fasta" while enabling edgeR abundance estimation and Trinity assembly. Subsequently, the mapping tables were merged, and an expression matrix was constructed. This allowed for differential expression analysis with the 'run_DE_analysis.pl' script. Finally, transcripts that exhibited highly significant differential expression (adjusted $p < 0.001$) and at least a four-fold ($\log_2$) difference at the isoform level were extracted and clustered. The results of this analysis can be viewed as a heatmap showing the differentially expressed genes (DEGs) in the samples. The DEGs in the heatmap were partitioned into gene clusters with similar expression patterns by cutting hierarchically clustered genes at a 60% cutoff (Ptree 60). This was performed using the script "define_clusters_by cutting_tree.pl". The expression matrix for each cluster ($\log_2$ transformed, median-centered) can be viewed as a line graph.

**Reference-based transcriptome assembly**

The raw data were first quality checked using FastQC v0.12.1 (*Andrews, 2010*); the data were filtered to remove the adapter sequences (provided by GenomeRead Inc.) using Fastp, and low-quality reads were removed using Trimmomatic v0.39 (*Bolger, Lohse & Usadel, 2014*). The "align_and_estimate_abundance.pl" script included in Trinity software (version 2.15.0, *Grabherr et al., 2011*) was used to map the *C. japonicum* reads to the *C. rubrum* reference transcriptome (*Pratlong et al., 2015*) using Bowtie2 (*Langmead et al., 2009*) for alignment and to calculate sample-specific abundance for each transcript using RNA-Seq by expectation maximization (RSEM) method (*Li & Dewey, 2011*).

The RNA-Seq power was calculated using an online implementation tool available at https://rodrigo-arcoverde.shinyapps.io/rnaseq_power_calc/. The calculation used the following parameters: sequencing depth of 30, sample size of 12, coefficient of variation of 0.37, effect size of 2, and significance level (α) of 0.001. The "run_DE_analysis.pl" script was used to determine the DEGs between sample groups (*i.e.,* female, male and immature) using the edgeR method in Trinity (*Robinson, McCarthy & Smyth, 2010*). DEGs were determined by controlling the false discovery rate (FDR) using an adjusted *p*-value < 0.05. DEGs were identified after running a script that performed pairwise comparisons among each sample group based on histological observations from the stereomicroscope.

The "analyze_diff_expr.pl" script was used to extract the highly DEGs (*i.e.,* most significant FDR and fold changes) that have an adjusted *p*-value less than 0.001 ($p < 0.001$) and have at least 4-fold ($\log_2$) differential expression between sample groups. The same procedure, mirroring the steps used for the *de novo* assembled transcripts, was then performed to create a heatmap and expression clusters for the *C. japonicum*-derived contigs with reference to *C. rubrum*.

## Similarity analyses of the *de novo* transcriptome assembly

The filtered transcriptome data "transcriptome_filtered.fasta" were used to generate a gene-to-transcript map using the Trinity assembly tool. Subsequently, peptide prediction was accomplished using the TransDecoder tool with the same dataset, specifically training with the longest open reading frames. To find similarities, the Diamond tool was employed for two distinct tasks: searching for analogous protein sequences (blastp) within the UniProt Swissprot database and comparing the translated nucleotide sequences (protein sequences) of *C. japonicum* against a protein database that includes sequences from *C. rubrum*. To retain only the most significant hits, an *E*-value cutoff at 1.0E-3 was applied. The protein accession IDs were queried using UniProt to identify and retrieve information on the matched protein sequences, including details about their identity, molecular function, and biological processes with which they are associated. Comprehensive transcriptome annotation was performed using Trinotate. The outcomes of these analyses were stored in a Trinotate-provided SQLite database template and viewed using SQLiteUI v.4.1.1 software (developed by Daniel Kummer of Germany). In addition, megablast was performed to compare the *de novo* assembled transcriptome to the reference-based transcriptome to validate the quality and accuracy of the *de novo* assembly.

## Functional enrichment analysis of *C. japonicum* transcriptome

Gene annotation analysis was performed to annotate the structural and functional biological information of the sequences. Gene subsets were analyzed using the Basic Local Alignment Search Tool (BLAST) using megablast, as described by *Afgan et al. (2018)*, which finds very similar sequences against intraspecies or closely related species in the public database of the National Center for Biotechnology Information (NCBI; *Sayers et al., 2022*). Gene subsets from females and males were matched separately to the transcripts from *C. rubrum* with annotated gene ontology (GO) terms (*Pratlong et al., 2015*) to determine whether the genes of interest were related to specific biological functions. Functional enrichment analysis was

**Table 1  Summary of histological observation of the precious coral *Corallium japonicum*.**

| Sample ID | Sex | Maturity | Average gonad diameter (μm) |
|---|---|---|---|
| PC01 | Female | Mature | $93.64 \pm 23.96$ ($n = 7$) |
| PC02 | Female | Mature | $157.34 \pm 31.88$ ($n = 8$) |
| PC03 | Male | Mature | $56.11 \pm 10.72$ ($n = 5$) |
| PC04 | Male | Mature | $46.8 \pm 6.34$ ($n = 6$) |
| PC05–PC12 | * | Immature | |

Notes.

*Sex was not determined.

performed for each sex using the GoEnrichment tool of *Afgan et al. (2018)* to determine significantly enriched GO terms.

## RESULTS

### Histological observation of *Corallium japonicum* gonads

Under the light microscope, four of the twelve individuals displayed small circular structures (average $95.21 \pm 50.52$ μm) in siphonozooids, and they were initially identified as 'mature' (Table 1). Notable size differences were observed between the two pairs (Figs. 1A and 1B). Two of these 'mature' individuals (PC01 and PC02) demonstrated relatively larger circular structures ($93.64 \pm 23.96$ μm and $157.34 \pm 31.88$ μm on average, respectively; Table 1) than the other two, and they appeared yellowish. These structures were identified as the oocytes. An average of two oocytes were observed in each siphonozooid. A large dark inner circle was observed in the oocyte center, which appeared to be the nucleus (Fig. 1A). The other two 'mature' individuals (PC03 and PC04) exhibited much smaller circular structures ($56.11 \pm 10.72$ μm and $46.8 \pm 6.34$ μm, respectively; Table 1) than the oocytes (Fig. 1A), and these circular structures were relatively smaller than those of the siphonozooids (Fig. 1B). The circular structures, which resembled sperm cysts, were light in color, but had no inner circles. They had fine but unclear structures (Fig. 1B). TEM showed that the unclear structures in these individuals were composed of densely packed polygonal cells (Fig. 2). The polygonal structures appeared to be spermatocytes inside sperm cysts. In these samples, 1–2 sperm cysts per siphonozooid were observed. No structures were observed in one individual (PC11) because histological analysis was performed on a dead portion of the nubbin with no available tissue; thus, this sample was provisionally identified as immature. The remaining seven individuals were considered immature because their circular structures could not be distinguished under a stereomicroscope. Additionally, hollow-like remnants identified in gonads after the release of gametes were also not observed. The presence of these structures is characteristic of an individual in a spent stage of development. Therefore, we concluded that PC01 and PC02 were female, PC03 and PC04 were male, and the other individuals were immature (PC05–PC12).

### *Corallium japonicum* transcriptome

RNA samples submitted for sequencing have concentration ranging from 7.0–44.0 ng/μl and an RIN of 4.8–8.7. Two samples had RIN that was not determined (PC02 and PC10),

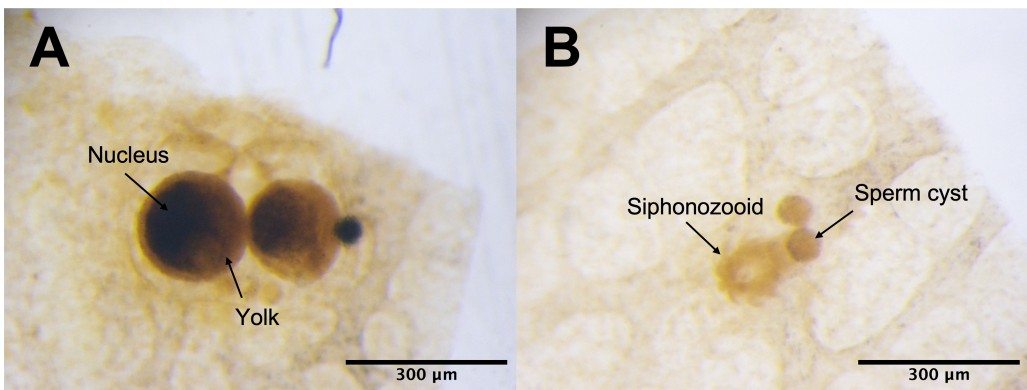

**Figure 1** ***Corallium japonicum* gonads observed under a stereomicroscope.** (A) Images of oocytes in PC02 showing the large dark nucleus and yellow yolk. (B) Images of light-colored sperm cysts in PC03 containing fine but unclear structures inside.

flagged as N/A by the bioanalyzer because of their low RNA concentration, which was below 10 ng/μl (Table S1). The overall range of the raw data received was 21–31 million reads. The statistical power of this experimental design, calculated using RNASeqPower, was 0.80.

## *De novo* assembled *C. japonicum* transcriptome

A total of 976,857 transcripts were initially assembled, of which 404,439 were retained after removing low-expression transcripts. The alignment rate varied between 70.92% and 77.69%, with an average rate of 74.89%. Among the filtered transcripts, 29,874 were common between females and males, 41,760 between females and immature individuals, and 42,126 between males and immature individuals. Of these 29, 874 transcripts, 855 exhibited significant differential expression ($p < 0.001$) at the isoform-level. Within this set, 786 transcripts displayed differential expression between females and males, one between females and immature individuals, and 175 between males and immature individuals (Fig. 3). The expression patterns of these 855 differentially expressed (DE) transcripts are visually depicted in a clustered heatmap (Fig. 4). In the heatmap, highly DE transcripts formed four subclusters with two major subclusters. Subcluster I included 283 upregulated transcripts (yellow bars) in two females and one immature individual (PC01(F), PC02(F), and PC10), which were downregulated (violet bars) in other samples. Subcluster II contained 525 transcripts that were primarily upregulated in males (PC03(M) and PC04(M)), with scattered yellow bars indicating moderate upregulation in immature individuals and downregulation in females. Subcluster III consisted of 46 transcripts that were highly upregulated in males and one immature individual (PC11), as shown by vibrant yellow bars. Subcluster IV contained one upregulated transcript in female individuals and some immature individuals (PC06–PC07 and PC10–PC12). Line graphs depicting the expression patterns for each subcluster are shown in Fig. S1.
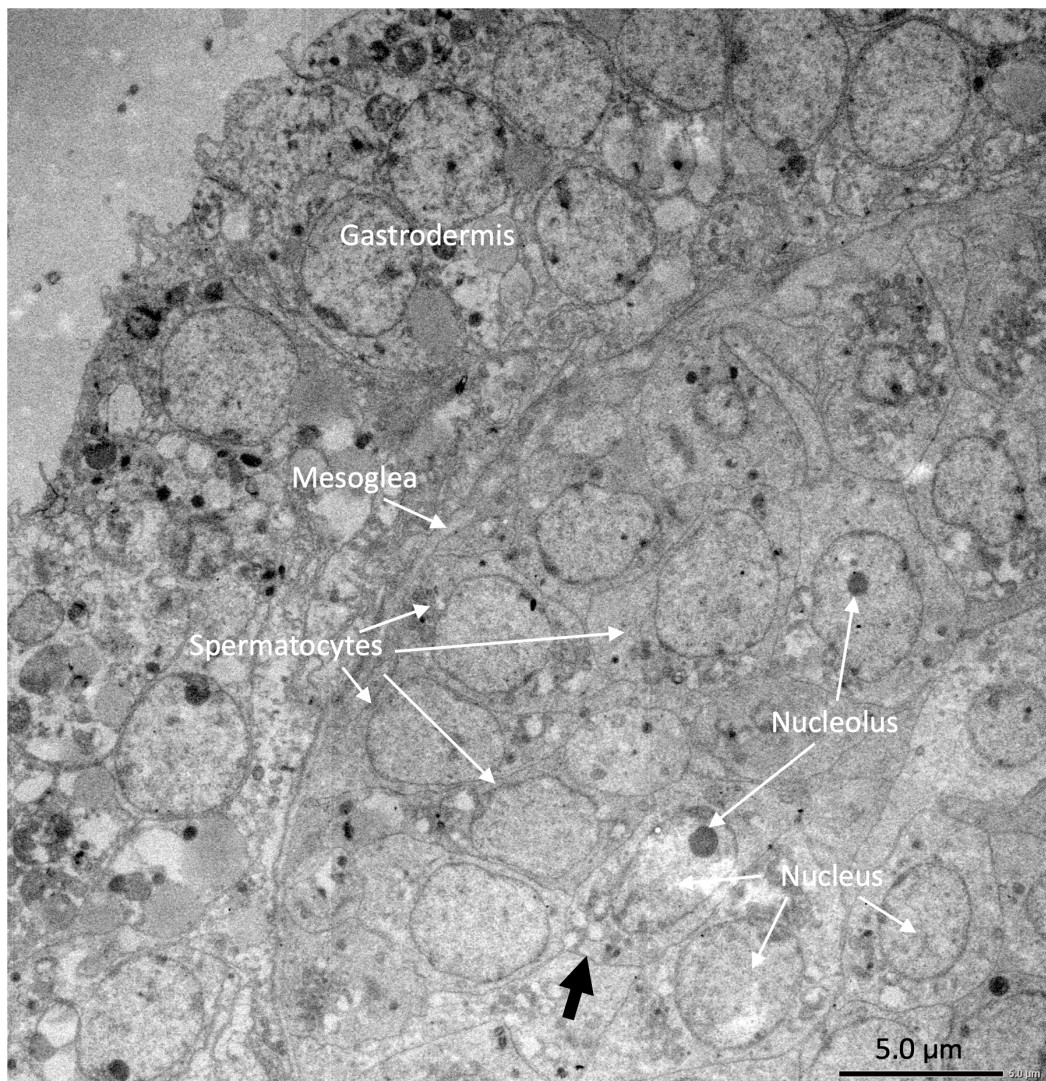

**Figure 2** **Electron micrographs of PC03 showing the sperm cysts containing the polygonal-shaped spermatocytes.** The spermatocytes are shown in a grey outline (pointed by the black arrow) which encloses a nucleus with a prominent nucleolus.

## Transcriptome assembled by mapping with *C. rubrum*

We observed that 60–65% of the short reads matched the reference transcriptome. A total of 28,092 contigs were assembled from these short reads. From these contigs, 693 DE genes ($p < 0.05$) were identified (Fig. S2A). Of these, 454 genes were DE between females and males, 300 were DE between females and immature individuals, and 99 were DE between males and immature individuals. Of the 693 DE genes, 206 were highly DE ($p < 0.001$). Of the 206 genes, 190 were DEGs highly expressed between females and males (Fig. S2B). The expression patterns of the highly DEGs showed sex-specific upregulation (Fig. S3), similar to the expression patterns of DE female and male transcripts assembled *de novo*.

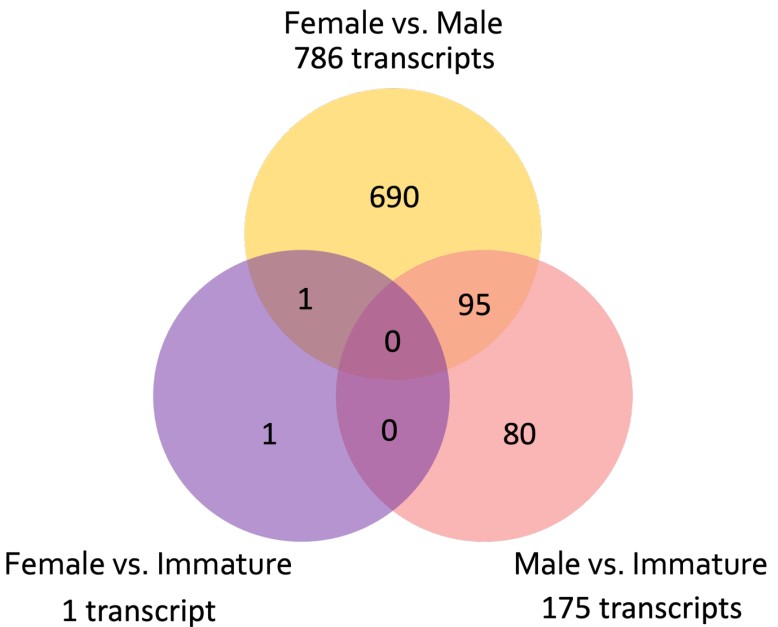

**Figure 3** Venn diagram showing the number of unique and common differentially expressed transcripts ($n = 855$; FDR > 0.001) across pairwise comparisons of sample groups: Female, Male, and Immature.

## Similarity analysis of the two transcriptome assemblies

The similarity search for DNA sequences (blastx) of *C. japonicum* transcripts assembled *de novo* and transcripts assembled after mapping with *C. rubrum* are summarized in Table 2, and the details are presented in Tables S2A and 2B. The summary table revealed that among females, 100 transcripts met the cutoff, with percentage identity (pident) ranging from 45.6% to 100% (average: 91.58%), alignment lengths from 22 to 971 bases (average: 296 ± 196), an average *E*-value of approximately 2.E-06, and bitscores ranging from 38.5–1,779 (average: 508.75 ± 357.61). In males, 63 transcripts passed the cutoff, with pident ranging from 40.8% to 100% (average: 82.58%), alignment lengths from 21 to 938 bases (average: 213 ± 196), an average *E*-value of approximately 2.E-07, and bitscores ranging from 43.1 to 1,790 (average: 327.81 ± 293).

Furthermore, we observed that some transcripts in the *de novo* assembly matched multiple contigs in the reference-based assembly (Table S3A). In the list of female DE transcripts, 74 contained sequences similar to 111 reference-based assembled contigs, whereas in males, 66 transcripts had sequences similar to 103 reference-based assembled contigs. These matches were influenced by a few transcripts that matched multiple contigs, ranging from two to six contigs per transcript in females and two to five contigs per transcript in males.

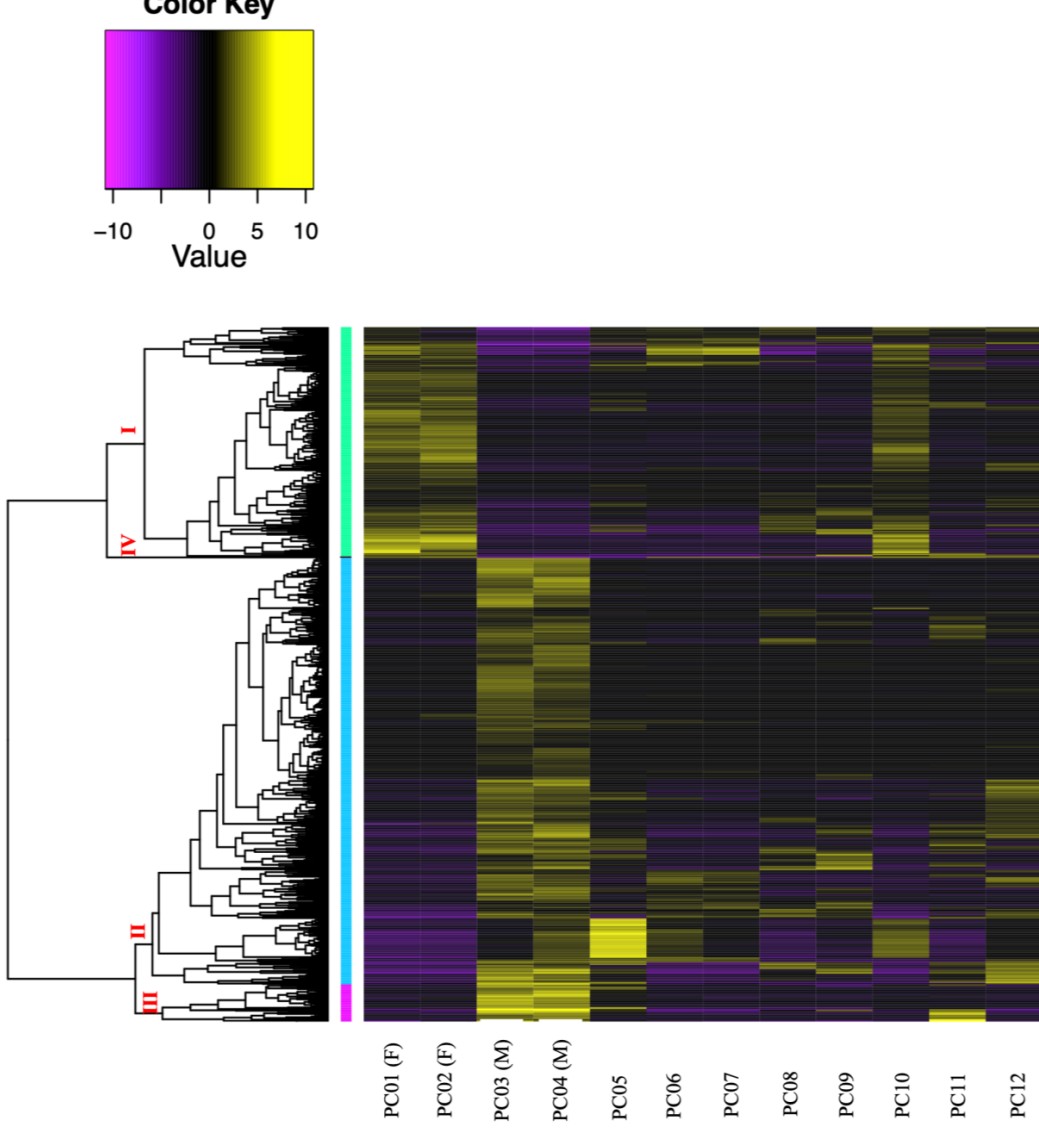

**Figure 4  Clustered heatmap (Ptree 60) of the *de novo* assembled differentially expressed transcripts (FDR > 0.001) showing the expression patterns of the three sample groups; female (PC01 and PC02), male (PC03 and PC04), and immature (PC05–PC12) in the six clusters.** Cluster numbers are shown in red font, Roman numerals. Transcripts belonging to subcluster I are grouped in light green vertical bars, while genes under subcluster II are grouped in light blue vertical bars.

## Functional enrichment analysis
### Annotation analysis of de novo assembled transcripts

Among the 976,857 assembled transcripts, only 155 (0.02%) had putative protein annotations. The same set of 155 transcripts accounted for 18.13% of the 855 DE transcripts. The results of the similarity search for analogous protein sequences (blastp) with significant hits (<0.001) for *de novo* assembled transcripts are summarized in Table 3 and the details can be found in Tables S4A and S4B. In females, 83 DE transcripts have

**Table 2** Summary of the similarity search for DNA sequences (blastx) between *de novo* assembled transcripts and mapping-based assembled transcripts of *Corallium japonicum*.

|  | pident | Length | Evalue | Bit score |
|---|---|---|---|---|
| **Female** | | | | |
| Min | 45.6 | 22 | 0.E+00 | 38.5 |
| Max | 100 | 971 | 2.E−04 | 1779 |
| Average | 91.58 | 296 | 2.E−06 | 508.57 |
| StDev | 12.42 | 196 | 2.E−05 | 358 |
| **Male** | | | | |
| Min | 40.8 | 21 | 0.E+00 | 43.1 |
| Max | 100 | 938 | 9.E−06 | 1790 |
| Average | 82.58 | 213 | 2.E−07 | 327.81 |
| StDev | 14.09 | 196 | 1.2E−06 | 292.85 |

**Table 3** Summary of the similarity search for analogous protein sequences (BLASTp) for *de novo* assembled transcripts.

|  | pident | Length | Evalue | Bit score |
|---|---|---|---|---|
| **Female** | | | | |
| Min | 22.6 | 17 | 2.E−208 | 41.6 |
| Max | 94.1 | 946 | 5.E−04 | 596 |
| Average | 49.3 | 227 | 2.E−05 | 168.9 |
| StDev | 16.4 | 191 | 8.E−05 | 127.8 |
| **Male** | | | | |
| Min | 23.0 | 46 | 7.E−134 | 46.6 |
| Max | 79.2 | 943 | 8.E−04 | 412.0 |
| Average | 41.0 | 261 | 4.E−05 | 150.9 |
| StDev | 13.9 | 218 | 2.E−04 | 110.9 |

matched putative protein annotation, with percent identity ranging from 22.6% to 94.1% (average: 49.48%), alignment lengths from 17 to 946 bases (average: $227 \pm 190$), E-values from 2.4E-208 to 4.6E-04 (average: $2.12E-05 \pm 8.E-05$), and bit scores from 41.6 to 596.9 (average: $168.87 \pm 127.20$). These proteins are involved in key cellular functions, including the cell cycle, division, metabolism, genetic processes, cell signaling, response, and structural dynamics. For males, 72 transcripts had significant hits revealing some of the putative protein annotation, showing pident ranging from 24% to 100% (average: 63.7%), alignment lengths from 33 to 1021 bases (average: $246 \pm 198$), E-values from 0E+00 to 5.36E-04 (average: $1.00E-05 \pm 6.51E-05$), and bitscores from 42.4 to 1087 (average: $277 \pm 224$). These proteins are primarily associated with genetic processes, metabolism and transport, cell structure and dynamics (including cell-to-cell adhesion), cell cycle and division, growth and development, and cell signaling and response.

**Table 4** Enriched GO terms in *Corallium japonicum*.

| Sex | GO ID | Term | FDR | Gene count |
|---|---|---|---|---|
| | GO:0007049 | Cell cycle | 1.E−08 | 20 |
| | GO:0051301 | Cell division | 2.E−06 | 11 |
| | GO:0000003 | Reproduction | 1.E−03 | 12 |
| Female | GO:0006996 | Organelle organization | 1.E−02 | 13 |
| | GO:0006807 | Nitrogen compound metabolic process | 2.E−02 | 43 |
| | GO:0008283 | Cell population proliferation | 2.E−02 | 8 |
| | GO:0043170 | Macromolecule metabolic process | 3.E−02 | 24 |
| Male | GO:0007155 | Cell adhesion | 3.E−02 | 4 |

## Comparative annotation analysis: *de novo vs.* reference-based transcripts

The annotations of the reference-based assembled transcripts coincided with the *de novo* assembled transcripts. Similarly, more female highly DEGs (84 of 119) have matched annotations in the NCBI public database of eukaryotic genes compared to males (49 of 87). GO term analysis revealed that 49 genes in females were enriched in seven GO terms, whereas four genes in males were enriched in one GO term (Table 4). In females, the enriched GO terms were cell cycle (GO:0007049), cell division (GO:0051301), reproduction (GO:0000003), organelle organization (GO:0006996), nitrogen compound metabolic processes (GO:0006807), cell population proliferation (GO:0008283), and macromolecular metabolic processes (GO:0043170) were the enriched GO terms. Both assemblies shared annotations for essential proteins, such as cyclins, cyclin-dependent kinases, histone proteins, and Vg1 mRNA (vitellogenin; Tables S4A and S5). These proteins play crucial roles in the seven specified biological processes (BP).

In males, cell adhesion (GO:0007155) was the most enriched BP. These enriched GO terms were consistent with the biological functions of the putative proteins annotated in the *de novo* assembled transcripts. Among the protein annotations, collagen alpha-5(VI) chain, focal adhesion kinase-1-like, and the GTP-binding protein ypt-1-like (Tables S4B and S6) were consistently upregulated in both transcript assemblies.

## DISCUSSION

### Histological observation of *Corallium japonicum* gonads

Similar structures were identified as oocytes and sperm cysts of the same species using the observation method described by *Sekida, Iwasaki & Okuda (2016)*. Our sample collection in early June aimed to capture completely mature individuals based on previous reports (*Nonaka, Nakamura & Muzik, 2015*; *Sekida, Iwasaki & Okuda, 2016*) and observations. Therefore, we expected to collect data from mostly mature individuals. In our initial identification, we categorized PC01- PC04 as 'mature' individuals. However, upon further comparison with previous studies (*Nonaka, Nakamura & Muzik, 2015*; *Sekida, Iwasaki & Okuda, 2016*), we discovered that our samples have relatively smaller gonad sizes. According to *Nonaka, Nakamura & Muzik (2015)*, mature oocytes of *C. japonicum* have a mean diameter of $362.1 \pm 10.4$ µm, whereas mature sperm cysts have a mean diameter of

222.2 ± 4.4 µm. Our oocyte specimens were 4× and 2× smaller than those used in their study, whereas the sperm cysts were 4× and 5× smaller (Table 1). These sizes aligned with the gamete developmental stages described in the same study. Our female samples were between Stages II and III and were characterized by oocytes surrounded by follicle cells with a conspicuous nucleus. In contrast, males approached Stage II, where spermatocytes were evenly distributed in the sperm cysts with no central open space, consistent with the results of our histological observations from the stereo microscope (Fig. 2). These findings suggest that the initially labeled 'mature' samples are still developing and are finally recognized as 'incompletely mature'. The term 'incompletely mature' will be consistently used henceforth. Thus, overall, only 33% of the samples were incompletely mature, and 67% were immature, potentially representing juvenile colonies. This suggests that the coral nubbins were collected slightly earlier. Considering the developmental stage of the gametes, it was estimated that PC01–PC04 still require 1–2 weeks to reach complete maturity and be spawned. Nevertheless, despite the incompletely mature stages of these four samples, we could distinguish between the females (PC01 and PC02) and males (PC03 and PC04).

Most samples (67%) comprised immature individuals, suggesting the presence of an additional cohort separate from the other two cohorts, representing incompletely mature female and male individuals. These cohorts are identified based on the size and appearance of their gonads. While sperm cysts mature annually, previous research on *C. japonicum* (*Nonaka, Nakamura & Muzik, 2015*; *Sekida, Iwasaki & Okuda, 2016*) has indicated multiple peaks in oocyte size, thereby suggesting a two-year reproductive cycle. Specifically, the cohort of mature oocytes spawned within a year, and the cohort of immature oocytes spawned the following year. This reproductive pattern was also observed in the oocytes of *P. elatius* (*Nonaka, Nakamura & Muzik, 2015*), *C. rubrum* (*Tsounis et al., 2006*; Vighi, 1972 in *Nonaka, Nakamura & Muzik, 2015*), *C. secundum* and *C. lauuense* (*Waller & Baco, 2007*). Consequently, if the immature individuals in our study are indeed female, it is expected that they would spawn in the following year. Subsequent transcriptome analysis focused on the gene expression of these four incompletely mature individuals (PC01–PC04).

### *Corallium japonicum* Transcriptome
### *De novo assembled C. japonicum transcriptome*
The *de novo* assembly of the *C. japonicum* transcriptome initially resulted in a large number of transcripts, 41.4% of which were retained after filtering for low-expression transcripts. A significant number of transcripts were common between sample groups (*i.e.,* females, males, and immature), suggesting a substantial overlap in the expressed transcripts. Significant differential expression (four-fold change, $p < 0.05$) was observed at the isoform level, with 855 transcripts exhibiting significant differences. Most of these transcripts were differentially expressed between females and males, highlighting the potential for sex-specific gene regulation. The expression patterns of these 855 DE transcripts are visually depicted in a clustered heatmap (Fig. 4), wherein 33% were upregulated in two females and one immature individual, and 61% were highly upregulated in males. In

summary, the *de novo* assembly of the *C. japonicum* transcriptome highlighted sex-specific gene upregulation during this stage of reproductive development.

Similar results were observed in a study by *Chiu et al. (2020)*, where *E. ancora* exhibited developmental and sex-specific gene expression variations, consistent with our findings during the middle stage of gametogenesis. In a study by *Van Etten et al. (2020)* on *Montipora capitata*, another Hexacorallia species, sex-specific gene upregulation was observed and sperm cells exhibited more upregulated genes (55%) than oocytes (45%). Our results align with *E. ancora* and *M. capitata*, wherein different genes were upregulated in each sex during the reproductive period of *C. japonicum*, which is an Octocorallia species. Moreover, one immature individual (PC10) showed an expression pattern similar to that of the incompletely mature females (PC01 (F) and PC02 (F)) (Fig. 4), suggesting that it represented a female at an earlier stage of development than PC01 and PC02.

Among the highly DE transcripts, most were differentially expressed between incompletely mature females and males. This was consistent with the results of our histological observations using stereomicroscope, wherein the same four samples (PC01 (F), PC02 (F), PC03 (M), and PC04 (M)) showed clear differences in appearance (Figs. 1A and 1B). Differential expression analysis also revealed a similar expression pattern in immature (PC10) and incompletely mature female individuals. Our analysis of the highly DEGs suggests that gene regulation is consistent among individuals at different stages in *C. japonicum*.

## Comparison of the two transcriptome assemblies

The transcripts assembled after mapping with *C. rubrum* (28,092) were significantly fewer than the *de novo* assembled transcripts (976,857), even after removing low-expression transcripts (404,439). The difference is likely due to variations where several contig sequences in the reference-based assembly matched a single transcript in the *de novo* assembly, and vice versa (Tables S2A and S2B). These variations stem from differences in how each assembly method reconstructs the genetic information, leading to discrepancies in the final transcript counts. These observations highlight the complexity of the transcriptome assembly process and the importance of using both *de novo* and reference-based approaches to gain a comprehensive understanding of *C. japonicum* gene expression.

Both *C. japonicum* and *C. rubrum* shared a subset of transcripts—100 in females and 63 in males (Tables S2A and 2B)—that exhibited significant similarity, with average percentage identity values of 91.58% in females and 82.58% in males. Moreover, both females and males showed higher average bitscores (508.75 and 327.8, respectively), indicating strong alignment with *C. rubrum*. These findings suggest that shared transcripts between the two species may represent consistently expressed constitutive genes, providing valuable insights into their functional roles.

Sex-specific gene upregulation observed in both assemblies suggests a genuine biological process, with the *de novo* assembly reducing potential biases. The consistency of sex-specific upregulation in transcripts mapped to the reference transcriptome further supports this observation. Additionally, comparisons of both female and male *C. japonicum* assemblies (Tables S2A and 2B) further validated the quality and accuracy of the *de novo* assembly.

The combination of these two transcriptome assemblies provides comprehensive insights into the gene expression patterns, facilitating cross-species comparisons and contributing to a deeper understanding of reproductive development in *C. japonicum*. Analysis of highly DE transcripts allows for the comparison of reproductive physiology between females and males at this specific developmental stage.

## Functional enrichment analysis

Our functional enrichment analysis provides valuable insights into the reproductive physiology of *C. japonicum* during gametogenesis. We found that only 0.01% of *de novo* assembled transcripts had matched protein annotations, a notably lower percentage than that reported in a previous study on *C. rubrum* (*Pratlong et al., 2015*). This disparity may be attributed to differences in annotation methods. Importantly, only 18.13% of the DE transcripts in our *de novo* assembly were annotated, whereas the reference-based assembly achieved a higher rate of 65%. This highlights the advantage of the reference-based approach in comprehensively annotating DEGs.

The putative protein annotations of female and male *C. japonicum* transcripts revealed distinct functional roles, which supports the sex-specific gene upregulation observed in the heatmap (Fig. 4). Female transcripts were associated with processes such as cell division, cell cycle regulation, mitosis, meiosis, and cell adhesion (Table S4A). Male annotations were primarily related to genetic processes, metabolism and transport, cell structure and dynamics (including cell adhesion), cell cycle and division, growth and development, and cell signaling and response (Table S4B).

Our analyses also revealed consistent findings for both assembly methods, confirming the upregulation of genes associated with cell proliferation and reproduction in females and cell adhesion in males, consistent with the typical activities of incompletely mature corals, as suggested by *Shikina et al. (2012)*. The expressions of histone proteins, cyclins, and cyclin-dependent kinases were upregulated, which may further validate our findings. These proteins play crucial roles in oocyte development and the accumulation of yolk proteins. After DNA replication in the oogonium, histone proteins integrate into the genome, followed by meiosis, transforming the oogonium into primary oocytes (*Raman et al., 2022*). This transition is governed by the formation of the Cyclin B and cyclin-dependent kinase-1 complex known as maturation-promoting factor (MPF), which promotes the developmental progression of oocytes through meiosis during oogenesis (*Marangos & Carroll, 2004*; *Feng & Thompson, 2018*). During oocyte development, they grow larger and accumulate yolk proteins (*Shikina et al., 2015*), corresponding to the enrichment of the BP term "Reproduction." The upregulation of vitellogenin, a major yolk protein precursor, suggests a potential role in oocyte accumulation for post-fertilization embryo nourishment, consistent with the observed yellowish yolk color in histological images (Fig. 1A), indicating that the oocytes were incompletely mature and still developing. Previous studies on *E. ancora* (*Shikina et al., 2013*; *Chiu et al., 2020*) support our findings regarding yolk accumulation during mid-stage oocyte development.

In males, the enrichment of cell adhesion and the upregulation of related proteins in both transcriptome assemblies of male *C. japonicum* suggest their relevance in the

current developmental stage. Cell adhesion molecules connect cells to the testes (*Li, Mruk & Cheng, 2013*; *Xiao, Mruk & Cheng, 2013*), support intercellular bridges, and facilitate synchronized development (*Greenbaum et al., 2006*). Our electron micrograph image (Fig. 2) revealed adjacent spermatocytes, indicating intercellular bridges, consistent with findings in the octocoral sea pen *Pennatula aculeata* (*Eckelbarger, Tyler & Langton, 1998*). Upregulated genes, such as focal adhesion kinase 1 (*Li, Mruk & Cheng, 2013*), GTP-binding protein (*Ke et al., 2018*), and collagen alpha-5(VI) chain (*Williams et al., 2021*), may play a role in maintaining and restructuring intercellular bridges. This process may contribute to synchronized spermatocyte differentiation into spermatids, which later mature into spermatozoa. As intercellular bridges dissolve, spermatozoa gain the ability to function independently, facilitating the fertilization of eggs. These findings suggest that the males were at a state of incomplete maturity, specifically the mid-stage of spermatogenesis.

Overall, the concurrence of analogous annotations for similar sequences in both *de novo* and reference-based assemblies confirmed reliable gene or transcript identification, reinforcing confidence in their accuracy and credibility. This consistency serves as an effective quality control measure, underlining the reliability of the assembly methods and the biological significance of these genes or transcripts.

## LIMITATIONS OF THE STUDY

Our study included information from only two female and two male individuals because the majority of the samples were in an earlier developmental stage, which made identification of sex challenging. The inaccurate estimation of the reproductive stage is partly attributed to the absence of data on the size at first maturity. Using this information, we can collect samples more accurately and increase the likelihood of obtaining mature individuals. The limitations stemming from our small sample size are evident in the clustered heatmap (Fig. 4), where genes expected to be upregulated only in males were also upregulated in one female individual (PC02F). Because we obtained only two sex-discriminating individuals of each sex, we could not perform statistical analysis (as the current standard for DE analysis requires at least three individuals per treatment). However, the upregulation of genes associated with the male sex in PC02F may also have been influenced by its low RIN value. Further investigation is required to clarify this. While the gene expressions of females and males from both transcriptome assemblies align with the results of histological analysis, the functional enrichment analysis remains inconclusive and requires further validation.

### Restoration of *C. japonicum* and other Japanese precious corals

Precious corals, which have a long history of exploitation, are now in need of active restoration. To address this pressing issue, recent restoration initiatives, such as the one by *Koido et al. (2022)*, have introduced innovative approaches like attaching coral fragments to artificial substrates using marine epoxy on land and deploying them to former precious coral habitats. Additionally, *Villechanoux et al. (2022)* have contributed to restoration efforts by testing various transplantation techniques across the Mediterranean, including the use of newly settled larvae on PVC tiles. Exploring a faster method to identify coral sexes shows promise for improved artificial cultivation and transplantation.
Using quantitative polymerase chain reaction to target genes identified in each sex within our study, valuable data on gene expression levels can be obtained, enabling individual sex determination and facilitating artificial cultivation. Successful *ex situ* breeding holds promise by enabling a balanced mix of sexes for transplantation, thereby improving the chances of fertilization. It may also provide insights into size at first maturity, addressing a research gap in Japanese precious corals.

## CONCLUSION

Our integrated analysis of histological observations and transcriptomes (both from *de novo* assembly and by mapping with *C. rubrum*) revealed that both female and male *C. japonicum* were in the mid-stage of gametogenesis, suggesting incomplete maturity. Although the limited sample size hinders conclusive findings, our results offer valuable insights into the physiological processes during this developmental stage. Females exhibited oocyte proliferation and reproduction, while males showed crucial cell adhesion for spermatozoa maturation. This study provides initial insight into the reproductive physiology of *C. japonicum* during gametogenesis. Further refinement of the sampling protocol and result analysis are essential for robust conclusions. The success and applicability of these methods may vary between different species and research conditions. To further validate the gene expression patterns of mature individuals, we recommend collecting samples from the same fishing field during July or August, when mature individuals are expected to be present. The use of sex determination methods based on gene expression differences has significant implications for conservation efforts, including transplantation, artificial fertilization, and larval rearing to preserve precious coral resources and ecosystems.

## ACKNOWLEDGEMENTS

The authors thank N Yoshimoto, H Kiuchi, M Miyamoto, K Kondo, K Kiuchi, S Kataoka, and R Kawamura for rearing and providing the precious corals in the aquarium tanks used in the experiments.

### Funding

This study was funded by the core research project "GAIA project - Utilization of Sustainable Marine Resources" and the doctoral research support program at Kochi University, and donations from the NGO The Precious Coral Protection and Development Association. The funders had no role in study design, data collection and analysis, decision to publish, or preparation of the manuscript.

### Grant Disclosures

The following grant information was disclosed by the authors:
Kochi University, and donations from the NGO.
Donations from the NGO.
The Precious Coral Protection and Development Association.

## Competing Interests

The authors declare there are no competing interests.

## Author Contributions

- Ma. Marivic Capitle Pepino performed the experiments, analyzed the data, prepared figures and/or tables, authored or reviewed drafts of the article, and approved the final draft.
- Sam Edward Manalili analyzed the data, prepared figures and/or tables, authored or reviewed drafts of the article, and approved the final draft.
- Satoko Sekida performed the experiments, analyzed the data, authored or reviewed drafts of the article, and approved the final draft.
- Takuma Mezaki performed the experiments, authored or reviewed drafts of the article, and approved the final draft.
- Tomoyo Okumura conceived and designed the experiments, authored or reviewed drafts of the article, and approved the final draft.
- Satoshi Kubota conceived and designed the experiments, authored or reviewed drafts of the article, and approved the final draft.

## DNA Deposition

The following information was supplied regarding the deposition of DNA sequences:

The sequences are available at NCBI SRA: PRJNA985876.

## Data Availability

The raw data is available in the Supplemental Files.

## Supplemental Information

Supplemental information for this article can be found online at http://dx.doi.org/10.7717/peerj.17182#supplemental-information.

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
