# Peer review of "Gene expression profiles of Japanese precious coral Corallium japonicum during gametogenesis"

_PeerJ, doi:10.7717/peerj.17182_

## Round 0.1 · original submission · Major Revisions

I have received evaluations from two expert reviewers and their comments can be seen below. Both reviewers recommend major revisions and I agree with their suggestions for improvement of the manuscript. Please ensure that you attend to all of the comments in a rebuttal letter where you clearly identify where the changes have been made.

·

Basic reporting

In this manuscript, entitled “Gene expression profiles of Japanese precious coral Corallium japonicum during gametogenesis”, Pepino and collaborators “performed a transcriptome analysis of the red Japanese precious coral Corallium japonicum to demonstrate the differences in gene expression between females and males during the spawning season” (l.97-99).
The Authors involved their study in a restoration ecology framework arguing the “transcriptome analysis is another method for identifying sex” (l. 80) of colonies to be used for transplantation (l.64-69).
To reach their objective, the Authors combined histological observations and transcriptomics of 12 colonies of C. japonicum.
Regarding histological observations, the Authors identified: i) mature vs immature individuals and within mature individuals, males vs females with a stereomicroscope (l. 135-139). The structure of the gonads was then described using transmission electron microscopy and gonade diameters were measured in sexed individuals (l;139-144).
Regarding the transcriptomic analyses, RNA samples were prepared and sequenced for all the 12 individuals (l.146-168). Differential gene expression analysis was conducted using standard methods (l.170-198) and followed by gene annotation and gene ontology analyses to “determine whether the genes of interest were related to specific biological functions” (l.206-207).
Histological observations supported that four of the twelve samples were matures with two males and two females. One sample was dead (l.213-231).
The Authors identified 693 differentially expressed genes (l.237) of which 206 were highly differentially expressed (l.240-241) with 119 in females and 97 in males (l.241-250).
Fifty-three of these genes matched with GO terms: 49 in females in seven GO terms and 4 in males in one GO term (l. 259-279).
Based on previous results in the same species, the Authors argued “the observed mature individuals were still undergoing development and were, therefore, incompletely mature” (l.290-291), speculated “that coral nubbins were collected slightly earlier” (l.299-300) and suggested “the presence of an additional cohort separate from the two cohorts, representing incompletely mature female and male individuals” (l.305-307) to explain the presence of immature individuals.
The proportion of DE genes was comparable to previously published study in E. ancora (l.317-326). Moreover, “at the current stage of gametogenesis, each sex exhibits specific gene
upregulation » as observed in other coral species (l.328-344). The Authors linked the pattern of DE with histological observations (l.346-352). They discussed the link between “the enriched BPs and six genes (i.e., histone, cyclins, cyclin-dependent kinase 1, vitellogenin, the focal adhesion kinase 1 and GTP-binding protein)” and the reproductive physiology of C. japonicum (l. 354-399).

Overall this is a well written manuscript reporting preliminary results in a poorly studied but interesting taxon owing to its economic value. The structure of the paper is appropriate and the submission is self-contained. Yet, various studies focused on another precious coral (C. rubrum) characterizing the reproductive cycle (e.g. Torrents & Garrabou 2011 Mar. Biol.) or using molecular sexing (Pratlong et al. 2017 Roy Soc Open Science) may be considered to improve the framework the study and the discussion of the results.
Moreover, the manuscript should most likely be re-oriented as a preliminary study owing to the very restricted sampling scheme (see below) and some missing information on the samples (e.g. size of the colonies?). In my opinion, the design of the study precludes robust conclusions (or at least question the validity of the findings) even if the two approaches (histological vs. RNAseq) point towards the same direction.

In this context, I would recommend the paper for publication with major revisions. Below I list some issues that should be addressed by the Authors.

Experimental design

The number of samples is relatively low compared to current standard. This is true for the histological observations and for the RNA sequencing. The low number of samples can be explained to some point considering the habitat of the species (deep habitat). Yet, even in this context, some information is missing. The size of the colonies sampled for the study is missing. Is there any data regarding the size at first reproduction in C. japonicum?
There is no mention of the number of histological slides prepared per individual, number of gonads encountered in mature individuals.
Regarding the RNA sequencing, there are only 2 males and 2 females when current standard for DE analyses are around 3 to 5 individuals per treatments (low bound). The Authors did not account for the implications of this very limited sample size on the robustness of their conclusions.
The limitations linked to the small sample size are, in my opinion, well-illustrated with the clustered heatmap on figure 4. The cluster green cluster which is supposed to include male upregulated gene clearly show yellowish horizontal bars in PC02F which is a female.
The Authors are most likely not able to provide more data (i.e. more males and females RNAseq). Accordingly, they should consider the potential caveats of their study and discuss their results accordingly. Otherwise, different parts of their discussions may appear a bit speculative (e.g. reproductive physiology l.362-399).

Validity of the findings

no comment

Additional comments

The introduction of the paper is mostly dedicated to the need to develop tools to support the restoration of precious corals. Yet, this topic is not considered in the discussion. How does the present study contribute to the restoration of C. japonicum?

The Authors mentioned that most of the individuals were immature. Yet, I was wondering how did they differentiate between immature individuals and individuals that already released their gametes?

Current knowledge regarding the reproductive cycle of C. japonicum (size at first reproduction?) or related precious corals from Corallium genus should be better explained in the Introduction. For instance, the occurrence of a bi-annual cycle in C. rubrum females vs. annual cycle in C. rubrum males has been previously published (Torrents & Garrabou 2011 MarBiol).

The abstract is sometimes difficult to follow and should be rephrased and enhanced.

Reviewer 2 ·

Basic reporting

In this study, the authors examined reproduction in Japanese precious coral Corallium japonicum, by using tissue section and transcriptome analysis. The methods and findings are valid and novel and therefore worthy of acceptance in this journal. However, as there are many unclear points in the analysis method, I would like to see it revised in light of some of the following points before acceptance.

Experimental design

From reading the methodology, I assume the authors also used the already reported assembled transcriptome of C. rubrum (from Pratlong et al., 2015), but did the authors assemble it using raw data from Corallium japonicum? I was unclear on the details around this, so I would recommend assembling the raw data from C. japonicum and then preparing a C. japonicum-derived contig with reference to C. rubrum before analysing it. If you have already done so, I would appreciate it if the authors could describe it so that it is clear.

Validity of the findings

The findings are valid and novel and therefore worthy of acceptance in this journal.

Additional comments

Lines 67 and 72: "Hirose & Hidaka" is a paper on the observation of gonads in coral bleaching and is not suitable for citation here.

Line 79: "said"->"sex"

Line 80: I think this sentence could be supplemented with an explanation of why transcriptome analysis is necessary." However, transcriptome analysis is another method for identifying sex." -> "Transcriptome analysis is another method for identifying sex because. ###."

Lines 127-129: How many colonies were used in the experiment: 12 colonies?

Line 147: "The samples were thawed" -> "The frozen samples were thawed".

Line 172: I recommend that the authors show the DRA number too.

Lines 176-178: "The "align_and_estimate_abundance.pl" script included in Trinity software (version 2.15.0, Grabherr et al., 2011) was used to map the reads to the C. rubrum reference transcriptome (from Pratlong et al., 2015)" was not clear. If the authors want to quantify expression, I think it would be better to map to the contig of C. japonicum itself.

Line 186: Between which treatments was the edgeR DEG extraction performed, between P01-02 and P03-04?

Lines 236-237: "From these short reads, a total of 28,092 contigs were assembled" was not clear. Wasn't it mapped to the originally assembled transcriptome (from Pratlong et al., 2015)?

Lines 318-319: What does "log2" mean?

Lines 322-326: As mentioned above, I do not know the details of the contig of Corallium japonicum, so we cannot interpret this sentence.

Lines 406-408: RNAseq may be new, but "the first-ever description of precious coral's reproductive physiology" may be not so.

It would be helpful if the RNA integrity number (RIN) was also given in the results.

---

## Round 0.2 · Major Revisions

I have received evaluations from the two reviewers who have reassessed the new version of the manuscript. Both agree that many issues have been attended to, however, both also comment that there are points of concern and in general the paper is hard to follow; especially with respect to the new sections that have been added. Please ensure that you take into account all of the reviewers´ comments and suggestions for improving the manuscript.

·

Basic reporting

The manuscript is sometimes hard to follow with some repetitions and confounding use of terminology (e.g. partially mature, immature, mature).
The hypothesis in particular those to explain the different results between the de novo vs. reference transcriptome approaches are sometimes unclear and speculative (e.g. focusing on the sampling season to explain the difference between japonicum and rubrum transcriptome).

Experimental design

Please see additional comments

Validity of the findings

Please see additional comments

Additional comments

This is the second time I review the manuscript entitled “Gene expression profiles of Japanese precious coral Corallium japonicum during gametogenesis” by Pepino and collaborators. I acknowledge the rephrasing and re-analyzing work conducted by the Authors in order to address the different comments made during the first round of review as well as the addition of a paragraph dedicated to the limitations of their study.
Yet, in my opinion, the manuscript is still not publishable in its present state. The Discussion section is quite long and should be significantly improved. In particular, the paragraph comparing the results obtained from the de novo vs. reference transcriptomes (from l. 494-558) is confusing and highly repetitive. The reasoning behind the hypothesis made by the Authors are unclear. For instance, they discuss the differences in the season of sampling between their study and the study that published the reference transcriptome in C. rubrum. In my opinion, they should focus on the fact the two species are sharing similar transcript (i.e. comparison among species) rather than discussing a potential effect of season or sex on their results. Below I list some issues to be addressed.

The Authors also mentioned that most of their samples are immature (e.g. l.429). I was wondering whether they were able to distinguish between juvenile colonies and maturing adult colonies. If yes, this should be clearly exposed in the material and methods for instance. If not, I would recommend to consider somewhere in the main text that “non-mature colonies” can instead potentially be juvenile colonies.

In this context, I recommend to reject the manuscript with possibility to submit an enhance version.


Some of the concerns related to the Discussion section:
From l. 423-444: please identify the part of the results discussed coming from stereo, vs. electron microscope. The sentence “These findings were consistent with the results of our histological observations (Fig. 1).” is confusing since all the part is based on histological observations.

l.447: “other two cohorts”. How were these two cohorts identify?

l.450: “non-annual two-year reproductive cycle” can likely be simplified as “two-year reproductive cycle”

l.454-455: “Therefore, if the immature individuals in our study were female, they were expected to spawn the following year.” This is unclear, please rephrase

l.456: four incompletely mature individuals. Please homogenize the use of “mature” vs. “incompletely mature” individuals.

l.471: please delete “underscoring the significance of these findings.”

l.487-488 is a repeat from l. 479-482. Please rephrase.

l.489-492: “Analysis of GO enrichment and annotations of the highly DE genes suggested commonalities in gene regulation among individuals at different stages in C. japonicum. This highlights the complexity of gene regulation in this species, warranting further research to uncover specific functions at different developmental stages.” Are the similitudes in gene expression patterns among individuals of the same species something surprising and suggesting some complexity? My guess is that this is expected and it is not clear to me how this suggests complexity. This should be rephrased and strengthened.

l.516: “highlighting the statistical significance biological relevance of these matches.” Unclear please rephrase.

l.536-543: This main idea of this paragraph is unclear. It appears mostly repetitive from previous paragraphs. Please rephrase or delete.

Reviewer 2 ·

Basic reporting

I could confirm that the authors have edited the manuscript in respect of the points I commented on previous manuscript. However, there are still some points of concern and I await the authors to revise and comment again.

Lines 236-237: "From these short reads, a total of 28,092 contigs were assembled" was not clear. Wasn't it mapped to the originally assembled transcriptome (from Pratlong et al., 2015)?
-> Yes, we utilized the C. rubrum transcriptome (from Pratlong et al., 2015) as a reference for mapping the sequence reads of C. japonicum to assemble our transcriptome. This approach allowed us to leverage the existing transcriptome information and align our reads to facilitate the assembly process.
>>
Sorry, I still wasn't clear here - do you mean that you referred to the transcriptome information of C. rubrum to get the annotation information of C. japonicum?

Lines 233-235: "Finally, transcripts that exhibited highly significant differential expression (p < 0.001) and at least a 4-fold (log2) difference at the isoform level were extracted and clustered. "
>>
I understand that edgeR was used to compare the three groups, but what is the ratio of the 4-fold to what? Also, does "p < 0.001" mean "FDR < 0.001"?

Lines 277, 370, 373, 392, 397: "evalue"->"E-value"

Lines 226-227
"among various sample groups, including females, males, and immature specimens."
->
"among three sample groups, namely, females, males, and immature specimens."

Experimental design

The authors have edited the manuscript in respect of the points I commented on previous manuscript.

Validity of the findings

The authors have edited the manuscript in respect of the points I commented on previous manuscript.

---

## Round 0.3 · Minor Revisions

We have received the comments back from one expert reviewer, who has now reviewed this manuscript three times and has pointed out that some changes need to be made. The reviewer and I agree that the manuscript requires a detailed reading and proofreading of the manuscript to ensure that it is of the quality that is acceptable for publication in PeerJ.

**Language Note:** The Academic Editor has identified that the English language must be improved. PeerJ can provide language editing services - please contact us at copyediting@peerj.com for pricing (be sure to provide your manuscript number and title). Alternatively, you should make your own arrangements to improve the language quality and provide details in your response letter. – PeerJ Staff

·

Basic reporting

This is the third time I review the manuscript entitled “Gene expression profiles of Japanese precious coral Corallium japonicum during gametogenesis” by Pepino and collaborators. I acknowledge the rephrasing and re-analyzing work conducted by the Authors in order to address the different comments made during the first two rounds of review.
There are still some parts that need rewording or rephrasing. For instance, l.502-505:
"These variations stem from differences in the definition and were defined and reconstructed by the two assembly methods, leading to discrepancies in the final transcript counts."

I recommend minor revisions and encourage the Authors to conduct an in-depth proofreading.

Experimental design

NA

Validity of the findings

NA

Additional comments

NA

---

## Round 0.4 · accepted · Accept

I sincerely appreciate that you have taken the time to proofread and ensure that the manuscript is of the standard required by PeerJ. I am satisfied that the manuscript can now be accepted for publication. Congratulations.